

# High-resolution in situ observations of atmospheric thermodynamics using dropsondes during the Organization of Tropical East Pacific Convection (OTREC) field campaign

Holger Vömel[1], Mack Goodstein[1], Laura Tudor[1], Jacquelyn Witte[1], Željka Fuchs-Stone[2], Stipo Sentić[2],
5  David Raymond[2], Jose Martinez-Claros[2], Ana Juračić[2], Vijit Maithel[3], and Justin W. Whitaker[4]

[1]National Center for Atmospheric Research, Boulder, CO, 30301, USA
[2]New Mexico Tech, Socorro, NM, 87801, USA
[3]University of Wisconsin, Madison, WI, 53706, USA
[4]Colorado State University, Fort Collins, CO, 80523, USA

10  *Correspondence to*: Holger Vömel (voemel@ucar.edu)

**Abstract.** The Organization of Tropical East Pacific Convection (OTREC) field campaign investigated the dynamical structure of convection in the tropical east Pacific and Caribbean. One of central data sets for this field campaign is the thermodynamic structure of the atmosphere measured by dropsondes released from the NSF/NCAR G-V research aircraft. Between 7 August and 2 October 2019, 648 dropsonde were successfully released from twenty-two research flights. Soundings were launched in 15  a grid pattern with a typical spacing of 1° longitude and 1.2° latitude and provide profiles of pressure, temperature, humidity, and winds between the surface and on average 13.3 km. Of these soundings, 636 provided complete vertical profiles of all parameters with a nominal vertical resolution between 6 to 12 m from the surface to almost flight altitude. OTREC deployed the new NRD41 dropsonde, which is the most advanced model that has been developed at NCAR. Here, we describe the data set, the processing of the measurements, and general statistics of all dropsonde observations. The dataset is available at 20  https://doi.org/10.26023/EHRT-TN96-9W04 (UCAR/NCAR, 2019).

## 1.   Introduction

The Organization of Tropical East Pacific Convection (OTREC) field campaign (Fuchs-Stone et al., 2020) was conducted to study the distribution of deep atmospheric convection in the tropical East Pacific. The main science objectives were to determine the distribution of deep convection in this region, including especially its day-to-day variability, why higher rainfall 25  rates occur over lower sea surface temperatures, and why easterly waves form and intensify in the Far East Pacific off the coasts of Central America and Colombia. To address these questions, OTREC conducted a two-month long field campaign of NSF/NCAR Gulfstream-V (G-V) aircraft observations using the NCAR Airborne Vertical Atmospheric Profiling System (AVAPS, NCAR 2020) dropsondes and the HIAPER Cloud Radar (HCR). In addition, ground-based remote sensing observations of integrated precipitable water vapor and balloon borne profiling at two sites in Costa Rica and one site in 30  Colombia were conducted. Here, we detail the sounding observations using the NCAR AVAPS dropsondes.



The NSF/NCAR G-V flew twenty-two research flights between 7 August and 2 October 2019, during which 648 dropsondes were successfully released. Two different flight track patterns had been defined prior to the campaign, originating in Liberia, Costa Rica, which are shown in Figure 1. Each nominal flight pattern consisted of eight legs with four sounding locations each for a total of 32 scheduled soundings per flight. The horizontal spacing was typically the equivalent of 1° longitude and 1.2°

latitude. The Caribbean pattern was rotated 45° following the general orientation of the coastline in that basin.

Only minor changes of the initially planned flight tracks happened for a variety of reasons. Research flight 6 on 18 August 2019 deviated from the regular pattern and operated in coordination with a simultaneous NOAA P-3 research flight out of Liberia. Research flight 17 on 25 September 2019 extended the Eastern Pacific lawn mower pattern to the south and skipped most of the Caribbean drop locations. The tracks for all 22 flights and the locations of all dropsonde releases are shown in

Figure 2.

Table 1 provides an overview of all dropsondes, which were released during OTREC. Some flights did not achieve the scheduled number of drops due to weather, failure of the dropsonde launcher, changes in the flight plan, or aircraft problems. Table 2 provides an overview of the performance of the entire dropsonde system. In total, 657 sondes were released from the aircraft. Nine soundings failed at launch and provided no data. In eight soundings, the telemetry stopped before the sonde

reached the ground. In four soundings, the GPS unit failed and provided no winds. All issues encountered are discussed in detail below. The overall success rate of the dropsonde system for this campaign is at 96.8% and demonstrates the high reliability of this observing system.

## 2.   AVAPS Dropsonde sounding system

The NCAR AVAPS dropsonde system deployed in OTREC used the automated dropsonde launcher on board the NSF/NCAR

G-V and the newly developed NCAR Research Dropsonde model NRD41. This dropsonde uses the pressure, temperature, and humidity sensor of the Vaisala RS41 radiosonde and employs an improved version of the GPS, telemetry, and parachute release system of the previous NRD94 dropsonde, which was in use between 2011 and 2018. It was successfully tested during the Southern Ocean Clouds, Radiation, Aerosol Transport Experimental Study (SOCRATES, McFarquhar et al., 2020) field campaign in January and February of 2018. OTREC was the first field campaign that relied entirely on this dropsonde model.

NCAR developed the smaller NRD41 dropsonde in parallel with its larger version, the RD41 dropsonde. This larger version has been introduced into operational service by NOAA and the Air Force in 2018 and is commercially produced and marketed by Vaisala. The reliability of the measurements can be considered equivalent between both types. The largest functional difference is the launch procedure and parachute release. While the larger RD41 is used exclusively in manual dropsonde launchers, the smaller NRD41 can be used in manual and automated dropsonde launchers. The NRD41 also uses a much more

reliable parachute release and launch detect mechanism, leading to far fewer launch detect and fast fall problems.

The NRD41 and RD41 (in short xRD41) dropsondes make use of the heated humidity sensor of the Vaisala sensor unit, which eliminates common bias and icing problems in humidity measurements. Temperature is measured by a platinum-wire sensor,



pressure is measured by a solid-state pressure transducer, and position and velocity are measured by Global Navigational
Satellite System (GNSS) positioning. Pressure measurements of the dropsondes were checked just prior to launch using a high

precision reference barometer installed inside the automated launcher.

The AVAPS LabVIEW based software (version 4.1.2) received and stored data from the dropsondes, the aircraft data system,
and controlled and monitored the AVAPS launch system.

The automated dropsonde launcher was installed in the baggage compartment of the NSF/NCAR G-V and remotely controlled
from the AVAPS station on board the aircraft. This allowed dropsonde operations up to a maximum altitude of 14.9 km, while

providing easy access to the launcher in case of malfunction.

Profile data were transmitted after the completion of each drop to the OTREC operations center at Playa Panama, Costa Rica,
where OTREC scientific staff controlled the quality of each sounding using the Atmospheric Sounding Processing
ENvironment (ASPEN) software package version 3.4.2 (https://www.eol.ucar.edu/content/aspen). The quality controlled data
of all soundings that did not raise any quality concerns were transmitted to the Global Telecommunications System (GTS) of

the WMO, which allowed data centers assimilating these data for analysis and forecasting.

During the first half of the campaign, a small number of sondes did not launch properly and became stuck in the launcher.
These sondes had to be removed manually, before new sondes could be loaded and released. The launch problems were
exacerbated during research flight (RF) 11 on 4 September 2019, when the launcher stopped releasing sondes, and the flight
had to be aborted prematurely after the release of only 21 sondes. Repair of the launcher led to a delay in the original flight

schedule. We speculate that between three and six of the failed dropsondes were damaged at release as a result of the launcher
damage. After its repair, the dropsonde launcher performed as expected and no further dropsonde release problems were
encountered.

## 3.   Quality control procedures

### 3.1.   Standard quality control

Standard quality control (QC) in near real time and as part of the final data QC is based on the algorithms implemented in the
ASPEN software. The following quality checks, corrections, and calculations are performed by ASPEN:

- Removal of outliers and suspect data points in pressure, temperature, humidity, zonal and meridional wind,
latitude, and longitude
- Removal of data between release from the aircraft and equilibration with atmospheric conditions
- Dynamic correction to account for the lag of the NRD41 temperature sensor using the appropriate coefficients
for the NRD41 dropsondes





- Dynamic correction to account for the sonde inertia in the determination of the wind profile using the appropriate parameters for the NRD41 dropsondes

- Smoothing of pressure, temperature, humidity, zonal and meridional wind
- Recomputing of wind speed and wind direction after smoothing of the wind components
- Extrapolation of the last reported pressure reading to a surface pressure value (where possible), based on the fall rate of the sonde
- Recalculation of the geopotential height from the surface to the top of the profile

- Computing a vertical wind speed component

During each flight, we processed each sounding as they were transmitted from the aircraft to the ground and generated the appropriate FM 37 TEMP DROP and 3 09 053 BUFR messages (WMO, 2020) using ASPEN. All data considered of high enough quality were sent to the WMO GTS for use in forecast and climate models.

The faster temperature and RH sensors required changing the ASPEN QC parameters for these two sensors, which were still set for the older NRD94 sondes. In particular, the equilibration time for the temperature and RH sensor was adjusted to 20 s, and the smoothing time for both parameters was adjusted to 5 s.

### 3.2. Additional quality control

All soundings were carefully investigated for any minor issue, which could not be handled by the standard QC using ASPEN.

None of the issues found were significant enough to send corrections to the GTS; however, they were corrected or flagged in the final set to provide the best possible data. The following sections describe the performance of the mechanical and measurement system components, and the relevant corrections applied that were not captured by ASPEN.

### 3.2.1. Pressure corrections

The pressure sensors used on board the NRD41 dropsondes are identical to those used in the Vaisala RS41 radiosondes. Unlike

the radiosondes, where a one point recalibration of the sensor is done prior to launch, here, the pressure sensor is recalibrated during the production of the dropsondes, leading to very low biases. In addition, the AVAPS dropsonde launcher included a Paroscientific reference pressure sensor, which measured the pressure inside the dropsonde launcher. This reference pressure was used to further reduce any residual bias of the NRD41 pressure sensor. The procedure was similar to how these pressure sensors are recalibrated prior to a radiosonde launch.

The statistics of the residual pressure bias measured inside the launcher is shown in Figure 3 and is based on the averaged difference over 30 s prior to launch for each sounding. The median pressure offset is 0.35 hPa and the standard deviation 0.17



hPa. Dropsonde pressure readings were corrected in post processing using the measurements for each sonde. The surface pressures reported by the dropsondes are expected to have only minimal systematic biases.

During OTREC, most sondes exhibited another small pressure measurement issue. For reasons unknown at the time, the
dropsondes occasionally repeated a reported pressure measurement. This happened up to 20 times per sounding and in a few cases more frequently. While this is barely noticeable in any vertical profile, it did cause additional noise in the calculated vertical fall rate. These repeated pressure readings were interpolated and the fall rates recalculated in post processing. Only pressure readings had to be corrected. Temperature and relative humidity readings did not show any artificial repetition of measurements. The source of the pressure repetition has meanwhile been attributed to a firmware bug inside the dropsondes.
The fix for this issue is currently under validation.

### 3.2.1. Temperature performance

The calibration of the temperature sensors was validated during production of the dropsondes and showed that their measurements agreed to within 0.15 K with a reference sensor under laboratory conditions with a two-sigma confidence level (k=2). During the campaign, all soundings but one showed consistent temperature observations within expected limits.

One sounding during RF17 (20190925_154412) shows a warm bias relative to its neighbors as well as to the nearby Nuqui radiosonde 100 km to the ESE, launched 30 min later. This bias varies between 1.0°C and 3.7°C throughout the profile. It also shows a significant geopotential height error relative to the other sondes. The temperature measurements of this instrument during the calibration check and prior to launch were within specifications, and we do not have any indication for a possible cause of this bias. Nevertheless, this sounding needs to be treated with caution.

With response times much less than 1 s, these profiles allow the highest vertical resolution of temperature measured by dropsondes.

### 3.2.2. Relative humidity

The calibration of the humidity sensors were validated during production at 75% relative humidity. The sensors agreed to within 3% with their reference (k=2). To achieve this level of confidence in atmospheric observations, the RH sensor on the
xRD41 dropsondes need to be reconditioned prior to launch to reduce the potential of sensor contamination to a minimum and to assure the best measurement performance throughout the entire altitude and temperature range of the profiles. The sondes store the information whether the reconditioning was successful and allowed us to verify that the operators properly reconditioned all sondes prior to take off before each flight. Any contamination in the sensor material was removed and the relative humidity sensors were expected to perform with negligible calibration drift.

The time response of the NRD41 relative humidity sensor is a few tenths of a second near the surface and nearly one minute at flight level of the G-V. A correction for this response time lag has not yet been implemented in ASPEN but was applied in





post processing following a similar approach as that by Kats et al. (2005). The effect of this correction was noticeable at altitudes above approximately 11.5 km and strongly increased the reported relative humidity near the top of the profiles.

Figure 4 shows the average relative humidity profiles for all OTREC soundings before the time lag correction (red) and after
time lag correction (blue). The effect of the time lag correction is significant only above 11.5 km, where the time constant of the sensor becomes very large, and where the reported profile shows a consistent vertical gradient. At 13 km, the time lag correction increased the relative humidity from an average value of 24% to 48%, i.e. by a factor of two. Ice saturation is at about 55% relative humidity (over liquid), which implies that the time lag corrected relative humidity measurements are more realistic for tropical measurements and closer to ice saturation, especially in regions where HCR observed clouds.
We removed the first 20 s of the relative humidity and temperature profiles, while the sensors were equilibrating to the ambient environment. Lacking any validating observations, some uncertainty in the relative humidity at the top of the profile remains and we would estimate that the layer 500 m below the aircraft should be treated with caution.

### 3.2.3. GPS performance

The GPS unit in the dropsondes operated properly in 95% of all soundings, i.e. the reported speed uncertainty of the GPS was
around 0.2 m/s in the lower part of the profile and around 0.4 m/s in the upper part of the profile.

Twenty-eight soundings (Table 3) had a slightly degraded performance with a speed uncertainty of 0.6 m/s in the lower part and up to 1.5 m/s in the upper part of the profile. ASPEN had been configured to remove the wind measurements under these conditions, which we noticed in the real time processing of these sounding. In post-processing, we increased the thresholds for the affected soundings, recovering the wind measurements that had been rejected in real time.
In three soundings (Table 4), the GPS module failed completely and no wind measurements were reported.

### 3.2.4. Launcher related problems

The launcher malfunction during the first half of the campaign led to damage in several sondes during the launch process. One indication of this damage was an internal sonde temperature much colder than normal. In four sondes, listed in Table 5, this
damage also led to a slower response of the atmospheric temperature sensor. The slower temperature response was noticeable only in the atmospheric equilibration after release, but not in the middle and lower troposphere. In these four profiles, we extended the equilibration time to remove any artefacts near the top of the profile.

Nine soundings completely failed at launch, i.e. either the telemetry stream stopped at launch, or the sondes reported a failure of the sensor modules at launch. Of these, between three and six may were damaged by the malfunctioning launcher. After
repair of the launcher, no further launcher related problems were observed and only one other sonde stopped working at launch.



### 3.2.5. Parachute performance

The parachute performed as expected in 98.2% of all soundings. In two sondes (Table 6), the parachute apparently did not function properly throughout the sounding and the sondes fell significantly faster than normal. The failure of the first sonde is likely related to the launcher malfunction. The failure of the second fast fall is less clear. In both cases, the estimation of the surface pressure may be low biased and the temperature profile may be slightly low biased as well.

Eight soundings (Table 7) experienced late parachute opening. In these soundings, the sonde was initially falling in an undefined orientation and the PTU measurements may have been negatively affected until the parachute properly opened. We removed these data where needed, to eliminate biased observations.

Sounding 20190822_172237 on RF07 experienced a slightly faster than normal fall rate down to 400 m above ground. All parameters are normal and the sounding was processed normally. Nevertheless, the parachute of this sounding may have been somewhat affected by the launcher malfunction.

Six soundings (Table 8) had a fall rate that was slightly but consistently slower than the expected fall rate. These sondes likely suffered some damage by the malfunctioning launcher, which increased the drag coefficient. This damage did not affect the GPS performance; however, it may have affected the performance of the temperature sensor in two soundings, in which the temperature equilibration took noticeably longer. Vertical velocities derived from these sondes should be treated with caution.

## 4. Sounding metrics

OTREC focused on tropical atmospheric dynamics and covered a two-month period with varying meteorological conditions. Here, we show summary figures and statistics to highlight the range of observations covered by the OTREC dropsonde data set. These summary figures also demonstrate the reliability of this dropsonde type.

Sondes were released at a aircraft speed of 235 m/s and a median altitude of 13.8 km (Figure 5). The drop altitudes at the beginning of each flight were typically at 13.1 km (43,000 ft); as the aircraft burned off fuel, the flight altitude increased up to a final ceiling altitude of typically 14.3 km (47,000 ft). Only one drop at the end of RF08 (20190823_173003) was released at a low altitude of 3.7 km (12,000 ft) for operational reasons. Dropsonde sensors require some time after release for equilibration to atmospheric conditions, which limits the effective ceiling altitude of the profiles roughly to 500 m below release altitude. Wind speeds during OTREC were typically less than 20 m/s in the upper part of the profile and less than 10 m/s in the middle and lower troposphere.. As a result, the horizontal drift of the dropsondes was relatively small (Figure 6). The mean horizontal distance the dropsondes traveled was 3.8 km and no sonde traveled more than 10 km horizontally. This horizontal drift is smaller than the horizontal resolution of most numerical weather prediction models.



The surface pressure reported by the sondes is an extrapolation of the last measured air pressure above the surface to sea level using the current fall rate. The surface pressure reported by all sondes, which transmitted to the surface, is shown in Figure 7. It varied typically between 1008 hPa and 1015 hPa. For most of the campaign, surface pressure variations do not reflect the

flight pattern, but rather slow changes of the larger scale meteorology.

A histogram of the measurement time for soundings with normal parachute performance is shown in Figure 8. Soundings with parachute failure and early telemetry loss are excluded from this plot. The average fall time for all soundings is 14.2 min. The increasing aircraft altitude during each flight contributes significantly to the width of the distribution. Nevertheless, the consistency of the fall times highlights the quality of the parachute performance of the dropsondes used in OTREC.

**5.  Atmospheric observations**

The temperature measured by all dropsondes is shown as contour plot in Figure 9. The temperature at flight level were in the range of -60°C to -70°C and near the surface in the range of 22°C to 29°C. The freezing level was on average at 4.9 km.

Relative humidity measured by all dropsondes is shown in Figure 10. At temperatures below 0°C, relative humidity is expressed as relative humidity over ice instead of the conventional relative humidity over liquid water. Areas near and above

ice saturation in the upper troposphere are periods when the aircraft flew in or above high level cirrus clouds.

Wind speeds are calculated by ASPEN based on the GPS horizontal velocity with a small correction for the intertia and the drag coefficient of the sonde (Lally and Leviton, 1958). Zonal wind speeds are shown in Figure 11. Brown colors indicate westerly winds, green and blue colors indicate easterly winds. Data files contain the wind components as well as wind speed and wind direction.

Data files include an estimate of the vertical wind speed, which is estimated based on the difference between the observed fall rate and the model fall rate for this sonde type (Wang et al., 2009). The vertical wind speed is calculated by Aspen and uses a drag coefficient of 0.52 and a sonde mass of 169 g. The calculation assumes that the parachute is working as expected and that the sonde geometry and mass are identical across all sondes. Therefore, the wind speed estimate calculated for fast-fall sondes (Table 6) and for partial fast-fall sondes (Table 7) is most likely incorrect. The vertical wind speed for sondes showing an

unusual fall rate (Table 8) is unreliable and should not be used either.

The overall estimate of the vertical wind speed uncertainty is less than 1 m/s. Areas of vertical updraft and downdraft above this uncertainty limit may be identified using this vertical wind speed estimate.

**6.  Data and code availability**

The OTREC dropsonde data are freely available at https://doi.org/10.26023/EHRT-TN96-9W04 (UCAR/NCAR, 2019). The

file format uses the Climate and Forecasting (CF) convention version 1.6 and is compatible with any tool accepting this

convention. The data file format is described in detail in Vömel et al. (2019), which follows that defined for the NCAR/EOL/ISF radiosonde NetCDF data files.

The ASPEN software package and a description of its functionality are available at https://www.eol.ucar.edu/content/aspen.

## 7. Conclusions

OTREC was an intensive campaign of measurements of the tropical atmospheric dynamics over the Eastern Pacific and Caribbean. The NSF/NCAR G-V aircraft flew 22 research flights during August and September 2019 and successfully released 648 NRD41 dropsondes, generating one of the largest dropsonde data sets from dedicated field campaigns. This was the first campaign that has seen extensive use of this dropsonde model. Its atmospheric measurements were of equal or better fidelity as those of the larger RD41 version of this sonde, which is used operationally by NOAA, the Air Force, and other organizations.

However, due to its smaller size and different parachute release mechanism, launch detect and parachute performance have been more reliable than those of the larger RD41 version.

Validation of the temperature and humidity sensor calibration during production provide a baseline for their performance, and recalibration of the pressure sensor using a reference just prior to launch eliminates almost all possible bias of the pressure sensor.

An additional quality control on top of the standard checks build into ASPEN was able to identify minor deviations from perfect behavior in a small number of soundings, which helps increasing the overall quality of the data.

The temperature, humidity, and wind fields measured by these sondes are being used for example to study details of tropical convection and provide a unique data set to investigate a multitude of other meteorological research questions.

## Author contributions


Holger Vömel analyzed all data, wrote the manuscript, and operated the dropsonde system. Mack Goodstein set up and configured AVAPS, and operated the dropsonde system. Laura Tudor operated the dropsonde system. Jacquelyn Witte handled the data archiving of the data. Željka Fuchs and David Raymond organized the campaign and acted as mission scientists. Stipo Sentić acted as mission scientist and performed real time data quality control. Jose Martinez-Claros, Ana Juracic, Vijit Maithel,
and Justin Whitaker performed real time data quality control.

## Competing interests

The authors declare that they have no conflict of interest.



**Acknowledgements**

The authors would like to express their appreciation for the skill and professionalism of the NCAR pilots and aircraft team,
who played a vital role in making this a successful campaign even in difficult situations.

**Financial Support**

This material is based upon work supported by the National Center for Atmospheric Research, which is a major facility
sponsored by the National Science Foundation (NSF) under Cooperative Agreement No. 1852977. Part of this work was
supported by the NSF grant 1758513 in support of field activities and post campaign analyses.

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

Meteorological Organization, WMO-No. 306, Geneva, Switzerland.




**Figures**

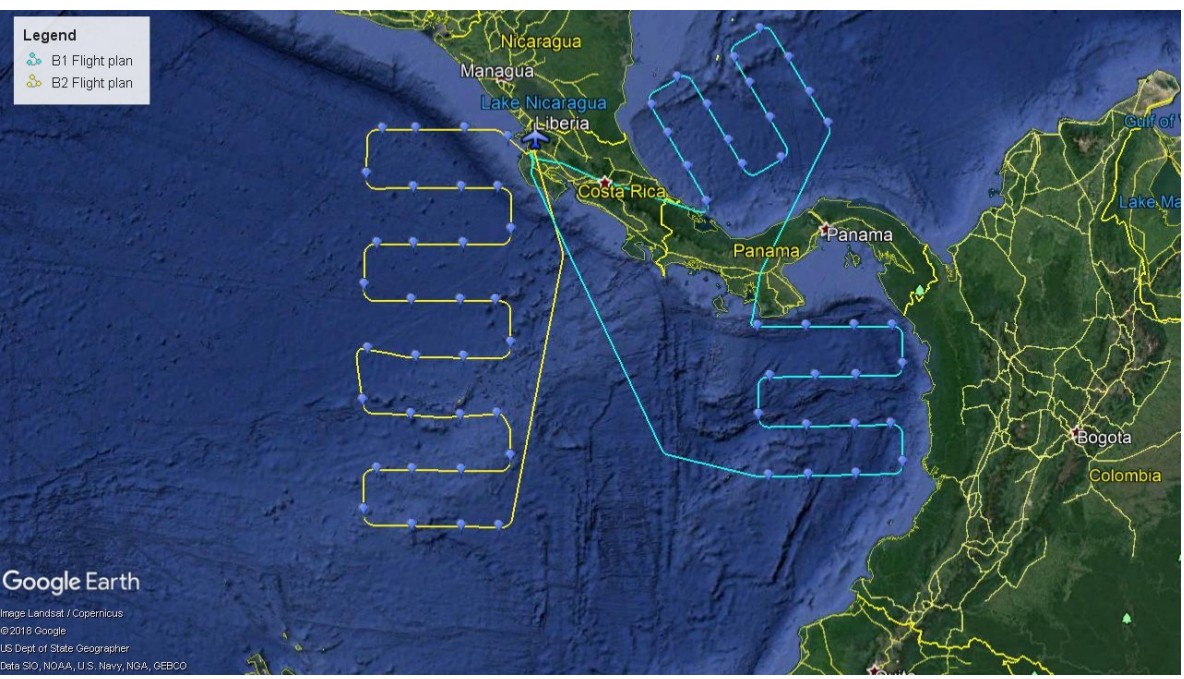

**Figure 1:** B1 (blue) and B2 (yellow) flight patterns and typical dropsonde locations.




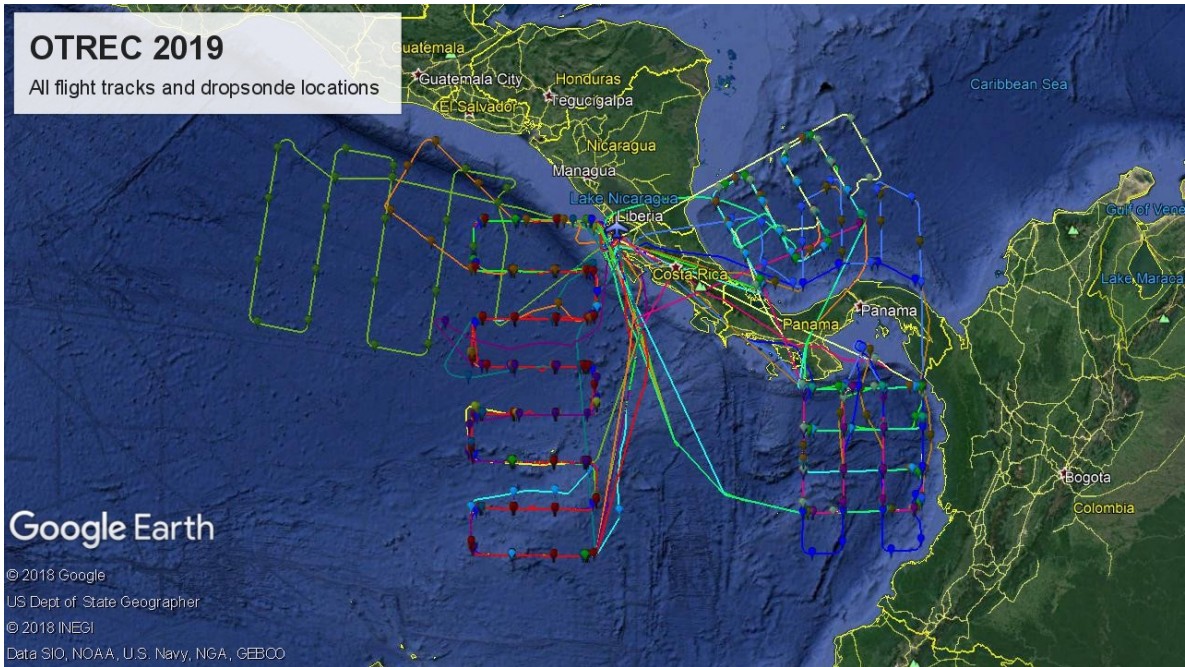

**Figure 2:** All flights tracks and all dropsonde locations during OTREC.




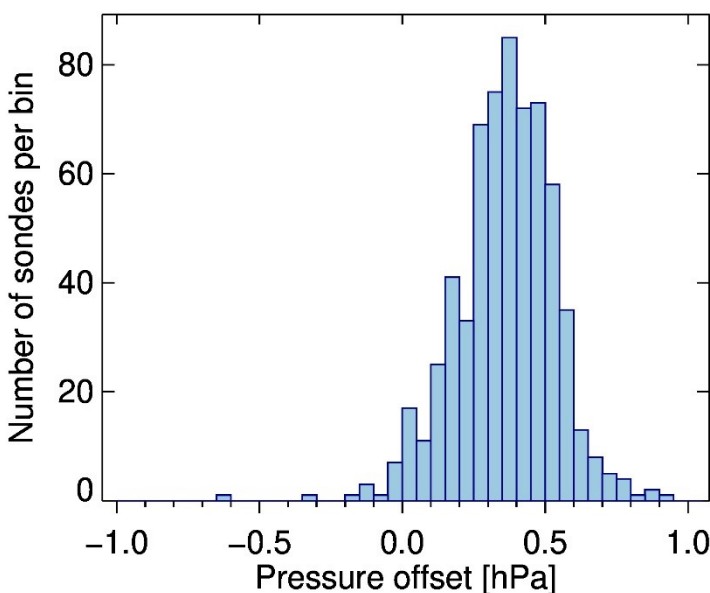

**Figure 3:** Pressure offset between the dropsonde and the reference sensor before launch.

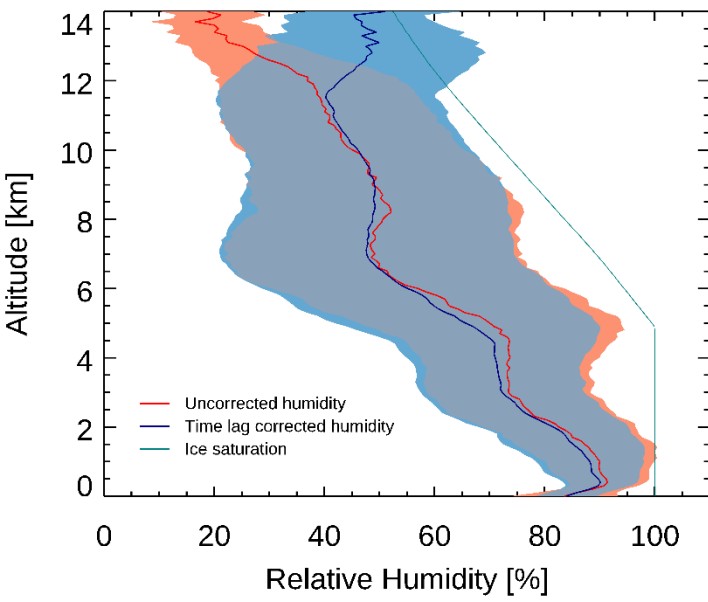

**Figure 4:** Mean relative humidity profile for all OTREC soundings. The average of the uncorrected relative humidity is shown in red, the average of the time lag corrected relative humidity is shown in blue. The standard deviation for each is shown as shaded areas. Saturation over ice is shown as solid green line.

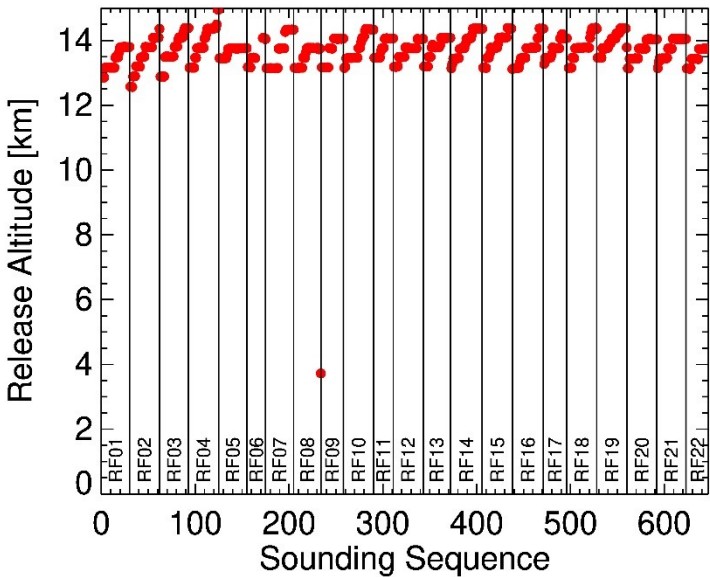

**Figure 5:** Time series of the release altitudes during OTREC. Sonde were typically launched above 12.1 km. Vertical lines separate the research flights, which are indicated near the bottom. The last sounding of RF08 was launched at low altitude due to operational reasons.





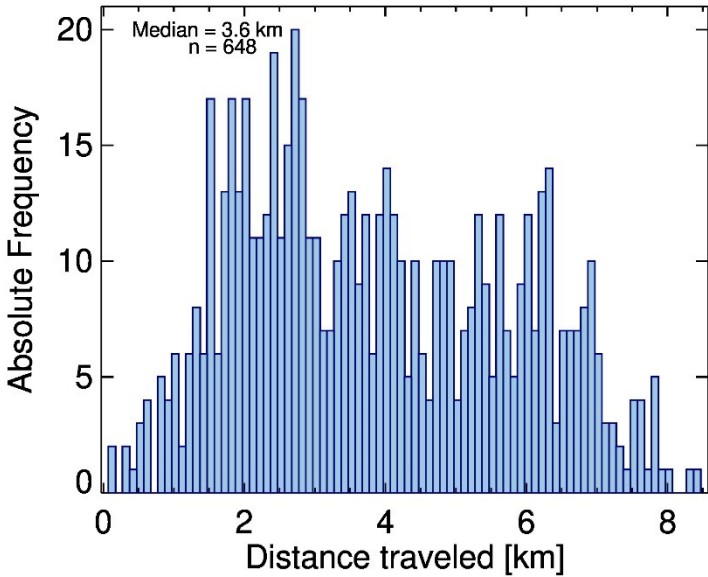

**Figure 6:** Distance between launch and landing for all dropsondes during OTREC.






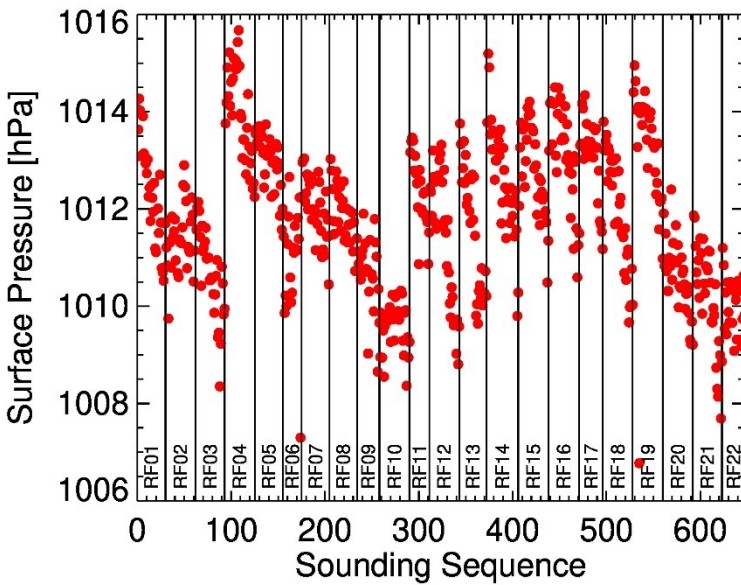

**Figure 7:** Surface pressure reported by all sondes. Vertical lines separate the research flights, which are indicated near the bottom.





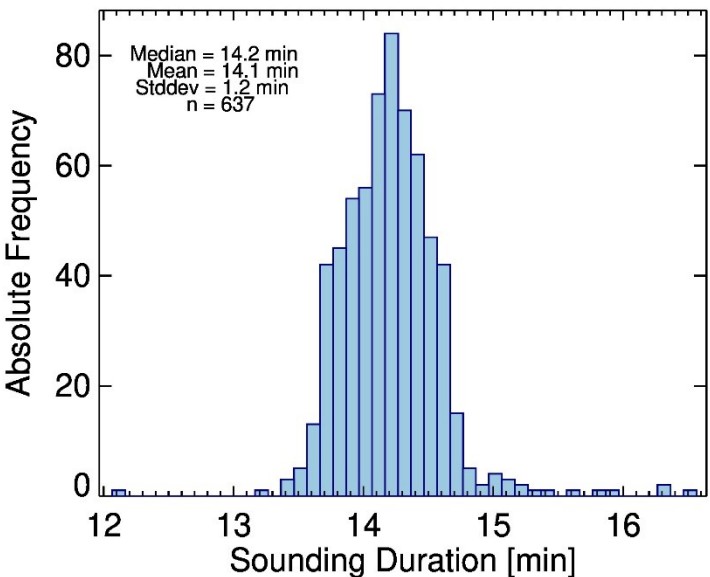

**Figure 8:** Measurement duration for all dropsonde with normal parachute behavior reaching the surface.

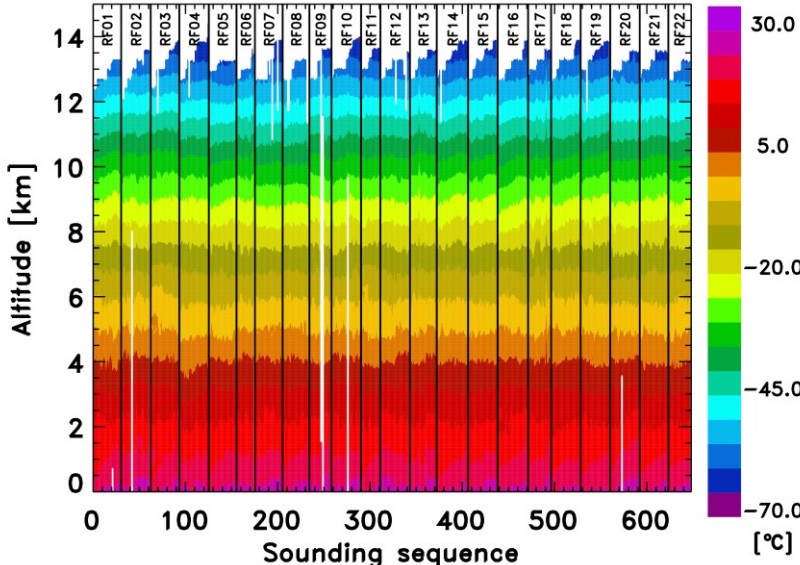

**Figure 9:** Color contours for all temperature measurements. Missing data are shown in white. All soundings are shown in the sequence in which they were released. Vertical lines separate the research flights, which are indicated in each group.


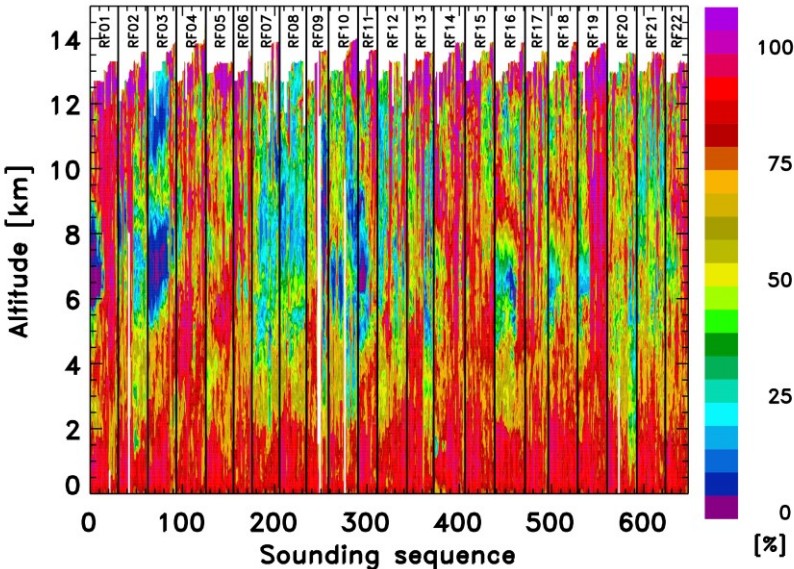

**Figure 10:** Color contours for all relative humidity measurements. Note that at temperatures below freezing, relative humidity is shown with respect to ice. Vertical lines separate the research flights, which are indicated in each group.



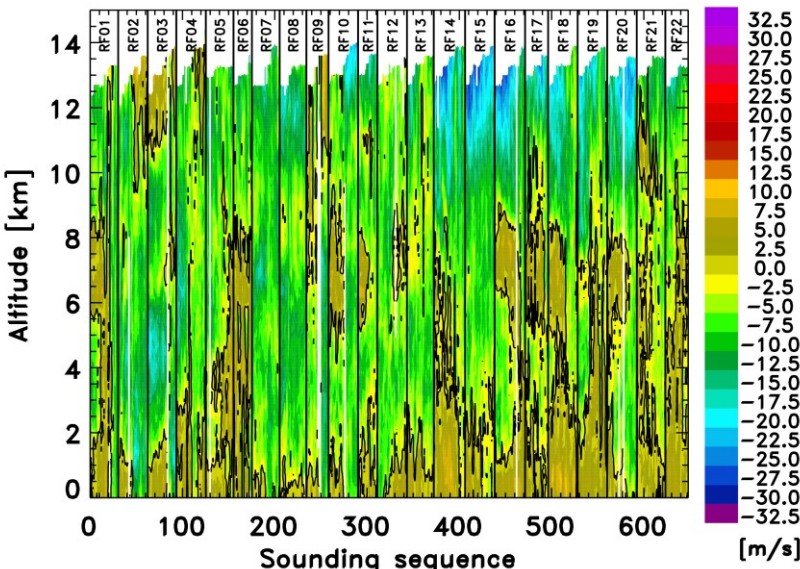


**Figure 11:** Color contours for all zonal wind speed measurements. The solid contour indicates the zero zonal-wind. Vertical lines separate the research flights, which are indicated in each group.





**Tables**

**Table 1:** Overview of all successful sonde releases during OTREC.

| Flight | Pattern | Date | # of Soundings |
|--------|---------|--------|----------------|
| RF01 | B2 | 07 Aug | 31 |
| RF02 | B1 | 11 Aug | 32 |
| RF03 | B2 | 12 Aug | 31 |
| RF04 | B1 | 16 Aug | 32 |
| RF05 | B2 | 17 Aug | 30 |
| RF06 | NOAA | 18 Aug | 20 |
| RF07 | B1 | 22 Aug | 30 |
| RF08 | B2 | 23 Aug | 29 |
| RF09 | B1 | 25 Aug | 24 |
| RF10 | B1 | 03 Sep | 32 |
| RF11 | B2 | 04 Sep | 21 |
| RF12 | B1 | 09 Sep | 32 |
| RF13 | B1 | 17 Sep | 29 |
| RF14 | B2 | 21 Sep | 34 |
| RF15 | B1 | 22 Sep | 32 |
| RF16 | B2 | 24 Sep | 33 |
| RF17 | B1 | 25 Sep | 25 |
| RF18 | B2 | 27 Sep | 32 |
| RF19 | B2 | 28 Sep | 32 |
| RF20 | B2 | 30 Sep | 32 |
| RF21 | B2 | 01 Oct | 31 |
| RF22 | B2 | 02 Oct | 24 |

**Table 2:** Overview of the dropsonde system performance.

| | # of Sondes | Percent |
|---|---|---|
| Total number of sondes released | 657 | 100 |
| Successful releases | 648 | 98.6 |
| Complete thermodynamic profiles to the ground | 640 | 97.4 |
| Complete wind and thermodynamic profiles to the ground | 636 | 96.8 |



**Table 3:** Soundings with degraded GPS performance. The speed uncertainty is reported by the GPS module.

*) In sounding 20190817_140439, the GPS altitude and GPS fall rate were completely wrong. Therefore, the horizontal winds were removed for the entire profile as well.

| # | Research Flight | Sounding | Median speed uncertainty [m/s] |
|---|---|---|---|
| 1 | RF01 | 20190807_132644 | 0.56 |
| 2 | RF01 | 20190807_145019 | 0.53 |
| 3 | RF01 | 20190807_154631 | 0.65 |
| 4 | RF01 | 20190807_160956 | 0.33 |
| 5 | RF01 | 20190807_171952 | 0.61 |
| 6 | RF02 | 20190811_160440 | 0.67 |
| 7 | RF02 | 20190811_164355 | 0.68 |
| 8 | RF04 | 20190816_150109 | 0.57 |
| 9 | RF04 | 20190816_170510 | 0.55 |
| 10 | RF05 | 20190817_140439 | 1.26* |
| 11 | RF05 | 20190817_162052 | 0.62 |
| 12 | RF06 | 20190818_141632 | 0.29 |
| 13 | RF06 | 20190818_175247 | 0.31 |
| 14 | RF07 | 20190822_183825 | 0.65 |
| 15 | RF07 | 20190822_184452 | 0.67 |
| 16 | RF08 | 20190823_140333 | 0.38 |
| 17 | RF08 | 20190823_154727 | 0.64 |
| 18 | RF11 | 20190904_141442 | 0.6 |
| 19 | RF12 | 20190909_162704 | 0.67 |
| 20 | RF12 | 20190909_165246 | 0.55 |
| 21 | RF12 | 20190909_174246 | 0.56 |
| 22 | RF13 | 20190917_155455 | 0.42 |
| 23 | RF14 | 20190921_142030 | 0.41 |
| 24 | RF14 | 20190921_175614 | 0.41 |
| 25 | RF15 | 20190922_145556 | 0.55 |
| 26 | RF15 | 20190922_154415 | 0.6 |
| 27 | RF16 | 20190924_165614 | 0.59 |
| 28 | RF19 | 20190928_153516 | 0.31 |



**Table 4:** Soundings, where the GPS module failed.

| # | Research Flight | Sounding |
|---|---|---|
| 1 | RF03 | 20190812_161306 |
| 2 | RF16 | 20190924_161135 |
| 3 | RF20 | 20190930_151914 |


**Table 5:** Soundings with slower equilibration after launch

| # | Research Flight | Sounding |
|---|---|---|
| 1 | RF03 | 20190812_142104 |
| 2 | RF07 | 20190822_172906 |
| 3 | RF07 | 20190822_181226 |
| 4 | RF08 | 20190823_164923 |


**Table 6:** Fast fall soundings

| # | Research Flight | Sounding |
|---|---|---|
| 1 | RF06 | 20190818_175247 |
| 2 | RF19 | 20190928_153516 |

**Table 7:** Partial fast fall and altitude of normal parachute performance

*) Temperature and relative humidity were set to missing above 11.7 km, GPS wind and altitude were set to missing above 12.5 km and additionally smoothed above 3.51 km .

| # | Research Flight | Sounding | Altitude of normal parachute operation [km] |
|---|---|---|---|
| 1 | RF04 | 20190816_150109 | 12.1 |
| 2 | RF07 | 20190822_163810 | 12.8 |
| 3 | RF08 | 20190823_140333 | 3.3[*] |
| 4 | RF08 | 20190823_141152 | 11.4 |
| 5 | RF12 | 20190909_171941 | 11.9 |
| 6 | RF12 | 20190909_182803 | 12.8 |
| 7 | RF12 | 20190909_184131 | 11.7 |
| 8 | RF14 | 20190921_135338 | 11.4 |



**Table 8:** Sondes falling slower than expected

| # | Research Flight | Sounding | Other symptoms |
|---|---|---|---|
| 1 | RF01 | 20190807_152431 | None |
| 2 | RF03 | 20190812_142104 | Slow temperature equilibration after launch above 11.6 km |
| 3 | RF06 | 20190818_175609 | Sonde data were lost prematurely at 8.5 km |
| 4 | RF07 | 20190822_172906 | Slow temperature equilibration after launch above 10.8 km |
| 5 | RF07 | 20190822_184452 | None |
| 6 | RF08 | 20190823_161835 | None |
