# Peer review of "High-resolution in situ observations of atmospheric thermodynamics using dropsondes during the Organization of Tropical East Pacific Convection (OTREC) field campaign"

_Earth System Science Data, 2020_

## Referee Comment (RC1) · Anonymous Referee #1 · 31 Dec 2020

General comments:

The manuscript presents the dataset of the dropsonde measurements carried out as part of the OTREC field campaign in 2019, which targeted deep convection regimes in the tropical east Pacific and in the Caribbean Sea, to study the dynamics and distribution of atmospheric deep convection. The authors provide an overview of these measurements including the sampling strategy, the details about the sounding system as well as the sonde instrument itself, quality control steps undertaken in real-time and during post-processing of the data, as well as a brief look into the relevant statistics

and metrics regarding the measurements.

The dataset is unique, as it provides a rich characterisation of the atmosphere in these deep convective regimes. The novelty is in terms of the number of sondes, as well as in the consistency of the sampling patterns. The diverse environments that the regions (East Pacific and Carribean) provide for deep convection to develop, make this dataset of considerable interest in understanding the thermodynamics as well as dynamics of deep convection and its interplay with the environmental parameters. Fuchs-Stone et al (2020) already give a glimpse of efforts in this direction. The colocated measurements from the cloud radar, as well as the ground-based measurements in the campaign can greatly benefit from the use of this dataset.

The files are all available as NetCDF (CF v1.6) files from the provided DOI, and are upto the highest standards one would expect from dropsonde data files. The files also provide QC parameters as additional attributes, which are definitely helpful in getting an idea of the performance of any sounding. The data collection (AVAPS and NRD41), quality control and the post-processing steps (ASPEN) are all state-of-the-art. The article also gives a good statistical overview of the success rate of measurements over several aspects, which can be checked from the files.

All that said, there are some minor clarifications and specifications that would certainly help the reader understand the data collection and QC better. I have specified these as notes below. I have also noted a few typographical and grammatical errors which the authors can use to help improve the draft.

Specific Comments:

Line 12: Typo. "One of central data sets..." Here a 'the' is missing between 'of' and 'central'

Line 14: Typo. "648 dropsonde were . . ." Should have been 'dropsondes' (plural).

Line 16: Grammar. "Of these soundings, 636 provided . . ." Tense consistency. The

previous sentence says 'provide' (present tense) in a similar context.

Line 28: Specify the type of ground-based measurements for IWV. I am guessing these are microwave measurements?

Line 60: Could there be a bit more elaboration on what is different in the parachute release between the NRD41 and the RD41? What makes the NRD41 more reliable? Or if there is some reference where this can be understood in greater detail, please cite it here.

Line 87: "suspect data points" Are the bounds for these suspect points the same as that are part of the standard editsonde configuration?

Line 95: Which type of smoothing (bspline, I think) and over what windows/wavelengths is this smoothing done?

Line 122: "... were corrected in post processing ..." Was this a simple offset correction applied to the whole profile? Or was the offset scaled with altitude? Additionally, the pressure sensor's uncertainty range (I checked for RS41 sensors) lies between 0.3 and 1 hPa (for 100 hPa and more). So to correct for a median offset of 0.35 hPa, there has to be a great confidence on the reference pressure sensor. What is the range of uncertainty for it? The model number of the same could be provided for the reader's reference.

Line 127: For clarification, did one repetition instance happen once or multiple times? i.e. Were there always pairs of the same value or sometimes also groups of the repeated value? In case it was the latter, what was the interpolation window? Also, I assume that all repeated values were discarded and then interpolation was carried out. Or can it be confirmed from the firmware bug which measurements were "true" measurements and which ones were repetitions?

Line 164: Does the uncertainty of the GPS module depend only on the satellite connections? Or something else too?

Line 165: Specify what is meant by upper and lower "parts" (e.g. halves, terciles, quartiles, 15%, etc.)

Line 177: For these 4 profiles, how much was the equilibriation time extended to?

Line 179: Grammar... "may have been"?

Line 188: Till the sonde came to its intended orientation, could it be that the equilibriation might also have been affected? And consequently, the equilibriation time needed to start after the parachute properly opens?

Line 209: Typo. Extra full stop.

Line 225: What about the deep convective clouds? e.g. from Fig. 10, soundings in some parts of RF01, RF19 and some others seem to show an atmospheric column that is almost completely saturated. Considering that HCR might have detected deep convection, would it be possible to mention in the manuscript how high were their tops and if the flight altitude was above the tops usually? Could these soundings be distinguished in some way from the ones in which cirrus were sampled close to flight level?

Line 226: Typo. 'inertia'

Line 228: Please include 'northward and eastward' before "wind components"

Figs. 1 & 2 : A scale is required. A lat/lon grid would be very much preferable. It would also be very helpful, if an additional scale shows the length of each leg and the gaps between each leg.

Fig. 6: This is only a recommendation. If the drift in sounding profiles can additionally be shown as deviation from launch locations (I have tried to plot it myself in figure attached), then one can also see that while the sondes did not drift much in the N-S direction, they did have a general east-ward drift. The current figure only provides displacement, but adding the drift in lat-lon can also give an idea of the direction of the

[Figure]

drift.

References:

Fuchs‐Stone, Ž., Raymond, D. J., & Sentić, S., 2020: OTREC2019: Convection over the East Pacific and Southwest Caribbean. Geophysical Research Letters, 47, e2020GL087564. https://doi.org/10.1029/2020GL087564

[Figure]

[Figure]

[Figure]

**Fig. 1.**

---

## Referee Comment (RC2) · James Franklin (Referee) · 4 Jan 2021

Review of ESSD-2020-325, "High-resolution in situ observations of atmospheric thermodynamics using dropsondes during the OTREC field campaign", by Vomel et al.

Reviewer: James Franklin

Recommendation and general comments: Accept with minor revisions. This is a very well-written and concise description of the new NRD41 dropwindsonde and the dataset collected for the OTREC field campaign. The contribution will be very useful to re-

searchers working with the OTREC data; indeed, because of its clear presentation of NCAR QC post-processing procedures, the paper will be useful to researchers working with just about any dropsonde dataset. I have only a few very minor comments and suggestions for improvements.

I'm not sure if this is ESSD style or the authors' personal style, but I found it difficult to identify paragraph breaks in the manuscript. With neither a blank line nor an indentation to mark the beginning of a new paragraph, I found myself frequently interrupting the flow of the reading to think about whether the authors were starting a new topic, particularly when encountering one-sentence paragraphs. I imagine other readers will have similar difficulty. Hopefully the ESSD house style allows for a more obvious identification of paragraph breaks.

The quality of the figures is generally good, although with figures 9-11 it's hard to tell exactly how the data values and colors correspond. For example, in Fig. 9 does the top-most purple color correspond to all data at least 30.0 but less than 35.0? Or are the colors centered on the listed values (27.5-32.5)?

Specific comments:

1. L25. In all my years as a hurricane researcher and forecaster, I wasn't aware of the argument or suggestion that easterly waves actually formed in the far eastern Pacific (or perhaps I've just forgotten in my advanced age). Could the authors please provide a reference?

2. L45. The sondes were all over-water releases, no? Maybe water, surface, or sea surface would be better choices than ground?

3. L103. I'm not sure what 3 09 053 refers to, but I assume that the listed reference would provide that information (I didn't check).

4. L107. I think it would provide helpful context to users to provide the equilibration times of the older sondes.

5. L110. I'm curious what would be the point of providing corrections to the GTS long after the data had been operationally ingested into numerical models.

6. L145. Data users might appreciate a little bit more information here on how the sensor contamination occurs and how the reconditioning process works.

7. L164. Can you describe how the reported speed uncertainty is determined and/or how a user should interpret it?

8. L185. If I recall, with the larger sonde fast fall data were not routinely transmitted for operational use due to concerns over data accuracy. I gather that you feel this is not an issue with the newer, faster sensors?

9. L209. Is this the total distance traveled, or the net distance between launch and splash?

10. Figure 3. Any speculation on why there was a positive bias?

Typos and editorial comments:

1. L14. ...648 dropsondes...

2. L102. ...each sounding as it was...

3. L179. ...between three and six were damaged...

---

## Referee Comment (RC3) · Sim Aberson (Referee) · 4 Jan 2021

High-resolution in situ observations of atmospheric thermodynamics using dropsondes during the Organization of Tropical East Pacific Convection (OTREC) field campaign by Vömel et al. is a concise and clear description of the dataset and of the instrument used to obtain the data. I recommend publication after one important and a few minor concerns are addressed:

Biggest concern: There is minimal mention of the data formats in which the dataset

is presented. There is also no mention of how problematic data as mentioned in Section 3.2 are noted in the dataset, if they are. It would be useful to have a little more explanation of this in Section 6.

1. Lines 87-88: It would be good to know some of the details of this removal: How does ASPEN define "outliers?" How does ASPEN define "suspect data points?"

2. Lines 91-94: Is a similar correction not necessary for moisture measurements?

3. Line 95: How is the smoothing done?

4. Line 100: Since this value has a direction, it should be velocity, not speed.

5. Line 117: Does paroscientific refer to the company, or something else?

6. Lines 120-124: How do these values correspond to the expected accuracy or the sensor? Are the values meaningful, or are they within the noise of the sensor?

7. Line 129: Does this issue impact sondes other than the NRD41, like the RD41? This would be important in looking at other dropwindsonde datasets.

8. Lines 136-137: How do you know that these data are erroneous, rather than just taken within a small-scale feature by chance?

9. Line 144: Should this read "within 3% RH" rather than 3%?

10. Line 179: Remove "may."

11. Line 259: "a multitude of other meteorological research questions" is mentioned, but it would be good to surmise as to what some of them might be.

---

## Author Comment (AC1) · 2 Feb 2021

**Replies to referee comments**

We would like to thank all referees for their careful review of our manuscript. We have considered all comments in the preparation of the revised version of the manuscript. Each comment is addressed below in detail. Our replies are indented to the comments raised.

**Anonymous Referee #1**

**General comments:**

The manuscript presents the dataset of the dropsonde measurements carried out as part of the OTREC field campaign in 2019, which targeted deep convection regimes in the tropical east Pacific and in the Caribbean Sea, to study the dynamics and distribution of atmospheric deep convection. The authors provide an overview of these measurements including the sampling strategy, the details about the sounding system as well as the sonde instrument itself, quality control steps undertaken in real-time and during post-processing of the data, as well as a brief look into the relevant statistics and metrics regarding the measurements.

The dataset is unique, as it provides a rich characterisation of the atmosphere in these deep convective regimes. The novelty is in terms of the number of sondes, as well as in the consistency of the sampling patterns. The diverse environments that the regions (East Pacific and Carribean) provide for deep convection to develop, make this dataset comment of considerable interest in understanding the thermodynamics as well as dynamics of deep convection and its interplay with the environmental parameters. Fuchs-Stone et al (2020) already give a glimpse of efforts in this direction. The colocated measurements from the cloud radar, as well as the ground-based measurements in the campaign can greatly benefit from the use of this dataset.

The files are all available as NetCDF (CF v1.6) files from the provided DOI, and are up to the highest standards one would expect from dropsonde data files. The files also provide QC parameters as additional attributes, which are definitely helpful in getting an idea of the performance of any sounding. The data collection (AVAPS and NRD41), quality control and the post-processing steps (ASPEN) are all state-of-the-art. The article also gives a good statistical overview of the success rate of measurements over several aspects, which can be checked from the files.

All that said, there are some minor clarifications and specifications that would certainly help the reader understand the data collection and QC better. I have specified these as notes below. I have also noted a few typographical and grammatical errors, which the authors can use to help improve the draft.

Specific Comments:

Line 12: Typo. "One of central data sets..." Here a 'the' is missing between 'of' and 'central'

Corrected.

Line 14: Typo. "648 dropsonde were ..." Should have been 'dropsondes' (plural).

Corrected.

Line 16: Grammar. "Of these soundings, 636 provided ..." Tense consistency. The previous sentence says 'provide' (present tense) in a similar context.

Changed to proper past tense.

Line 28: Specify the type of ground-based measurements for IWV. I am guessing these are microwave measurements?

These were actually GPS integrated water vapor measurements. We added the qualifier GPS.

Line 60: Could there be a bit more elaboration on what is different in the parachute release between the NRD41 and the RD41? What makes the NRD41 more reliable? Or if there is some reference where this can be understood in greater detail, please cite it here.

We changed that sentence and expanded a little: "In addition, the NRD41 uses a parachute release mechanism, which is electronically controlled and triggered after launch of the sonde. This method is more reliable than the mechanical delay ribbon used on the larger RD41, leading to far fewer launch detect and fast fall problems."

Line 87: "suspect data points" Are the bounds for these suspect points the same as that are part of the standard editsonde configuration?

Yes, we use the same filters for suspect data as are set in the editsonde configuration. The most important changes, i.e. the smoothing wavelengths were already noted in the manuscript.

Line 95: Which type of smoothing (bspline, I think) and over what windows/wavelengths is this smoothing done?

Correct, ASPEN uses a bspline algorithm. We added this information. For OTREC we used a final smoothing wavelength (time) of 5 s. We added this information as well; it is also contained in the metadata of the ncdf files.

Line 122: "... were corrected in post processing ..." Was this a simple offset correction applied to the whole profile? Or was the offset scaled with altitude? Additionally, the pressure sensor's uncertainty range (I checked for RS41 sensors) lies between 0.3 and 1 hPa (for 100 hPa and more). So to correct for a median offset of 0.35 hPa, there has to be a great confidence on the reference pressure sensor. What is the range of uncertainty for it? The model number of the same could be provided for the reader's reference.

We used a Paroscientific 6000 reference pressure sensor on the aircraft. The calibration of this sensor was checked in our own calibration lab before and after the campaign. The accuracy was within the specified range of 0.1 hPa. We added the model number in the manuscript.

Line 127: For clarification, did one repetition instance happen once or multiple times? i.e. Were there always pairs of the same value or sometimes also groups of the repeated value? In case it was the latter, what was the interpolation window? Also, I assume that all repeated values were discarded and then interpolation was carried out. Or can it be confirmed from the firmware bug which measurements were "true" measurements and which ones were repetitions?

When it happened, pressure values were repeated exactly once. The spacing between occurrences of these repetitions varied. The time series of the raw data made it clear that deleting the repeated value and interpolating between the two nearest neighbors is appropriate with minimal loss in accuracy. To clarify, we changed the word "repeated" to "duplicated".

Line 164: Does the uncertainty of the GPS module depend only on the satellite connections? Or something else too?

In addition to position and winds, the GPS modules used in the RD41 and NRD41 dropsondes provide an uncertainty of the reported speed. The algorithm used to calculate this uncertainty is protected by the manufacturer and we have no additional information what input parameters are used. The number of satellite signals received by the GPS unit is one of the essential parameters, but not there are others. One of which we are aware is the strength of the satellite signals. Other parameters may play a role, but we have no further information about these details.

Line 165: Specify what is meant by upper and lower "parts" (e.g. halves, terciles, quartiles, 15%, etc.)

We clarified this sentence and now state: "the reported speed uncertainty of the GPS was around 0.2 m/s below 10 km and around 0.4 m/s above."

Line 177: For these 4 profiles, how much was the equilibration time extended to?

We extended the equilibration time in these profiles to about 2 min, i.e. no temperature or humidity data above about 11.5 km are available. We added this time in the text.

Line 179: Grammar... "may have been"?

**Corrected.**

Line 188: Till the sonde came to its intended orientation, could it be that the equilibration might also have been affected? And consequently, the equilibration time needed to start after the parachute properly opens?

Especially in these high altitude drops, the launch causes a dramatic step change in pressure, temperature, humidity, and winds. The parachute opens about 3 s after launch, which is also when launch detect is set. This starts the equilibration correction in ASPEN. However, the orientation of the sonde in this early phase of the drop is not important for the sensors. The equilibration time has been set conservatively, to make sure that no sonde profile contains data that could be considered part of the equilibration period.

Line 209: Typo. Extra full stop.

**Corrected.**

Line 225: What about the deep convective clouds? e.g. from Fig. 10, soundings in some parts of RF01, RF19 and some others seem to show an atmospheric column that is almost completely saturated. Considering that HCR might have detected deep convection, would it be possible to mention in the manuscript how high were their tops and if the flight altitude was above the tops usually? Could these

soundings be distinguished in some way from the ones in which cirrus were sampled close to flight level?

While this is a very good question and something we have looked into, our manuscript focuses exclusively on the description of the dropsonde data set. Analyzing HCR and dropsondes together provides a wealth of information, in particular, since the humidity observations of the dropsondes extend higher with better reliability than any dataset before. In addition, the aircraft data include in situ measurements of particles, which could be included. We have not yet done this and may consider this in future manuscripts.

Line 226: Typo. 'inertia'

Corrected.

Line 228: Please include 'northward and eastward' before "wind components"

Corrected.

Figs. 1 & 2 : A scale is required. A lat/lon grid would be very much preferable. It would also be very helpful, if an additional scale shows the length of each leg and the gaps between each leg.

**We added a lat/lon grid as well as a length scale.**

Fig. 6: This is only a recommendation. If the drift in sounding profiles can additionally be shown as deviation from launch locations (I have tried to plot it myself in figure attached), then one can also see that while the sondes did not drift much in the NS direction, they did have a general east-ward drift. The current figure only provides displacement, but adding the drift in lat-lon can also give an idea of the direction of the drift.

The main point of this Figure was to demonstrate that the sonde profiles stay mostly within a single grid box in most but the highest resolving numerical weather models. We added a phrase that the majority of the drift was westward (not eastward).

**References:**

FuchsâA× RStone, Ž., Raymond, D. J., & Senti× c, S., 2020: OTREC2019: Convection′ over the East Pacific and Southwest Caribbean. Geophysical Research Letters, 47, e2020GL087564. https://doi.org/10.1029/2020GL087564 on Earth Syst. Sci. Data Discuss., https://doi.org/10.5194/essd-2020-325, 2020.

This reference was already included.

**Reviewer: James Franklin**

Recommendation and general comments: Accept with minor revisions. This is a very well-written and concise description of the new NRD41 dropwindsonde and the dataset collected for the OTREC field campaign. The contribution will be very useful to researchers working with the OTREC data; indeed,

because of its clear presentation of NCAR QC post-processing procedures, the paper will be useful to researchers working with just about any dropsonde dataset. I have only a few very minor comments and suggestions for improvements.

I'm not sure if this is ESSD style or the authors' personal style, but I found it difficult to identify paragraph breaks in the manuscript. With neither a blank line nor an indentation to mark the beginning of a new paragraph, I found myself frequently interrupting the flow of the reading to think about whether the authors were starting a new topic, particularly when encountering one-sentence paragraphs. I imagine other readers will have similar difficulty. Hopefully the ESSD house style allows for a more obvious identification of paragraph breaks.

Unfortunately, this is a feature of the ESSD template. The final typeset version will look better.

The quality of the figures is generally good, although with figures 9-11 it's hard to tell exactly how the data values and colors correspond. For example, in Fig. 9 does the top-most purple color correspond to all data at least 30.0 but less than 35.0? Or are the colors centered on the listed values (27.5-32.5)?

The temperatures next to the color bar give the lowest temperature of the color next to it. The top color indeed indicates the temperatures above  $+30^{\circ}$ C. Adding all temperatures next to the color bar makes it too crowded and we feel this a sufficient for these more qualitative plots. To help the reader, the text already specified that the lowest temperatures were in the range of  $22^{\circ}$ C to  $29^{\circ}$ C

Specific comments:

1. L25. In all my years as a hurricane researcher and forecaster, I wasn't aware of the argument or suggestion that easterly waves actually formed in the far eastern Pacific (or perhaps I've just forgotten in my advanced age). Could the authors please provide a reference?

We have modified this sentence to include several reference: "The main science objectives were to determine the distribution of deep convection in this region, including especially its day-to-day variability, and why higher rainfall rates occur over lower sea surface temperatures. Some research shows that strong easterly wave genesis might originate off the coast of Panama and Colombia in the latitude range of  $5^{\circ} - 10^{\circ}N$  (Kerns et al., 2008; Serra et al., 2010; Rydbeck and Maloney, 2014 and 2015)."

**The references are:**

Kerns, B., K. Greene and E. Zipser, 2008: Four years of tropical ERA-40 vorticity maxima tracks, Part I. Mon. Wea. Rev., 136, 4301-4319.

Serra, Y. L., G. N. Kiladis and K. I. Hodges, 2010: Tracking and mean structure of easterly waves over the Intra-Americas sea. J. Climate, 23, 4823-4840.

Rydbeck, A. V. and E. D. Maloney, 2014: Energetics of east Pacific easterly waves during intraseasonal events. J. Climate, 27, 7603-7621.

Rydbeck, A. V. and E. D. Maloney, 2015: On the convective coupling and moisture organization of east Pacific easterly waves. J. Atmos. Sci., 72, 3850-3870.

2. L45. The sondes were all over-water releases, no? Maybe water, surface, or sea surface would be better choices than ground?

Changed to "sea surface".

3. L103. I'm not sure what 3 09 053 refers to, but I assume that the listed reference would provide that information (I didn't check).

"3 09 053" is a designator for the WMO BUFR formats specifically defined for dropsonde observations. This is explained in detail in the WMO manual on codes given in the manuscript. Most importantly, this format allows transmission of high (i.e. 1 s) resolution data, in contrast to the lower resolved TEMPDROP format.

4. L107. I think it would provide helpful context to users to provide the equilibration times of the older sondes.

We added a sentence: "In particular, the response time of the temperature sensor is about 4 times faster than that of the older model."

5. L110. I'm curious what would be the point of providing corrections to the GTS long after the data had been operationally ingested into numerical models.

Even though sending corrected data will not affect the near real time forecast, we believe that corrected data may be used in reanalyses at some of the forecasting centers. However, we decided that this was not warranted here.

6. L145. Data users might appreciate a little bit more information here on how the sensor contamination occurs and how the reconditioning process works.

We added the following sentences: "Absorption of contaminants into the sensor material due to outgassing by packaging materials and other unidentified sources slowly degrades the calibration of the sensor. A dedicated heating cycle of the sensor prior to launch reconditions the sensor material and restores the original calibration. After successful reconditioning, the relative humidity sensors are expected to perform with negligible calibration drift."

7. L164. Can you describe how the reported speed uncertainty is determined and/or how a user should interpret it?

In addition to position and winds, the GPS module used in the RD41 and NRD41 dropsondes reports an uncertainty of the reported speed parameter. We have clarified, that this parameter comes directly from the GPS module. The uncertainty reported by the GPS is in units of m/s and gives an uncertainty of the horizontal speed measured by the GPS. The horizontal wind speed requires an additional dynamical correction, which accounts for the inertia of the sonde. Furthermore, smoothing of the speed measurements happens in the GPS unit (unquantified) as well as in the processing by ASPEN. While this smoothing may further reduce the uncertainty of the reported wind speed, it increases the thickness of the vertical layer for which this wind is reported. These details go beyond the scope of this paper and will be addressed in another manuscript. Here, we wanted to highlight that the uncertainty of the GPS speed measurements are less than 1 m/s, which is a typical threshold for high quality measurements defined by WMO. 8. L185. If I recall, with the larger sonde fast fall data were not routinely transmitted for operational use due to concerns over data accuracy. I gather that you feel this is not an issue with the newer, faster sensors?

In real time data QC, we tended to evaluate profiles conservatively and did not transmit profiles to the GTS if they were deemed suspicious, which included the few fast fall profiles. Most of these profiles passed post campaign final QC, where our goal was to provide as much data as possible. One of the purposes of this paper is to highlight potential biases, where they could be suspected. Having faster sensors obviously helps in fast fall profiles. The usefulness of fast fall data for use in real time analysis and forecasting is still to be decided. Having a dataset such as OTREC helps in addressing this question.

9. L209. Is this the total distance traveled, or the net distance between launch and splash?

We referred to the net distance between launch and landing in the water and added this in the manuscript.

10. Figure 3. Any speculation on why there was a positive bias?

At this point, we do not have an explanation. This is the first data set, where the pressure sensor correction was implemented in production AND checked prior to launch. We did expect a mean offset close to zero. We verified the calibration of the reference sensors used in production and in the aircraft. This result simply highlights the need for independent reference measurements of pressure prior to launch, which we are beginning to implement into AVAPS.

Typos and editorial comments:

1. L14. ...648 dropsondes...

Corrected.

2. L102. ...each sounding as it was...

Corrected.

3. L179. ... between three and six were damaged...

Corrected to " ... may have been damaged ... "

**Reviewer: Sim Aberson**

High-resolution in situ observations of atmospheric thermodynamics using dropsondes during the Organization of Tropical East Pacific Convection (OTREC) field campaign by Vömel et al. is a concise and clear description of the dataset and of the instrument used to obtain the data. I recommend publication after one important and a few minor concerns are addressed:

Biggest concern: There is minimal mention of the data formats in which the dataset is presented. There is also no mention of how problematic data as mentioned in Section 3.2 are noted in the dataset, if they are. It would be useful to have a little more explanation of this in Section 6.

We now explicitly state that the data files are in netCDF format and we provide a reference to the CF conventions. We have changed some text in the Data and Code Availability section to

"The files are in NetCDF format and use the Climate and Forecasting (http://cfconventions.org) metadata convention version 1.6. The file format and contents are described in detail in Vömel et al. (2019), which follows that defined for the NCAR/EOL/ISF radiosonde NetCDF data files."

At this point, the data files either contain valid or missing data. No quality flag has yet been defined and introduced into this file format that could be used to indicate questionable data. Our goal in the final data QC was to provide the largest amount of data without letting any data pass the QC that we judged as obviously bad.

We added the following sentences to the introduction to Section 3.2: "We also describe all occasions, were not all components of the instrument worked as intended. Data were set to missing if they showed obvious inconsistencies and were otherwise left in place."

1. Lines 87-88: It would be good to know some of the details of this removal: How does ASPEN define "outliers?" How does ASPEN define "suspect data points?" 2. Lines 91-94: Is a similar correction not necessary for moisture measurements?

The details of the ASPEN QC algorithms go beyond the scope of this paper. Here we would like to refer to the online documentation of ASPEN at

https://ncar.github.io/aspendocs/algo\_outlier.html

*Dr.* Aberson is correct, that a time lag correction is also required for the moisture measurements. However, these have not been implemented in ASPEN and were performed outside of ASPEN as described in Section 3.2.3.

3. Line 95: How is the smoothing done?

ASPEN uses a bspline algorithm. We added this information.

4. Line 100: Since this value has a direction, it should be velocity, not speed.

Corrected.

5. Line 117: Does paroscientific refer to the company, or something else?

*Yes, Paroscientific is the manufacturer of high quality reference pressure sensors. We added the model number in the manuscript.*

6. Lines 120-124: How do these values correspond to the expected accuracy or the sensor? Are the values meaningful, or are they within the noise of the sensor?

This is a very good question. The correction is larger than the noise of the senor and therefore meaningful. However, we do not yet fully understand the stability of the calibration. If the calibration were perfectly stable between manufacture and launch, then Figure 3 would have shown a very narrow distribution around 0. The shift and spread of the offset shown here indicates how stable the calibration was under the conditions the sondes experienced between manufacture and launch. The correction applied removes this bias and is therefore an essential step in getting the best possible pressure measurements. However, this test cannot address how the uncertainty of the pressure measurements is influenced by the launch and the dramatically changing conditions during a sounding. A true evaluation of the uncertainty of all sensors is in preparation.

7. Line 129: Does this issue impact sondes other than the NRD41, like the RD41? This would be important in looking at other dropwindsonde datasets.

This issue only affects the RD41 and NRD41, no older sonde models. A fix for this issue will be implemented in production soon. This issue is noticeable looking at the fall rate calculated from raw data. Affected profiles will show one point of slower followed by one point of faster fall rate. This may happen multiple times in a single profile. Any level of smoothing will reduce this artifact and effectively make it negligible for all but the highest vertical resolution studies.

8. Lines 136-137: How do you know that these data are erroneous, rather than just taken within a small-scale feature by chance?

All parameters indicate that in this profile the instrument performed as expected. However, out of 648, this profile shows the by far largest deviation from the mean temperature as well as the largest difference from its neighbors. At this point, we do not know if there is a geophysical reason for this unusual temperature structure. We decided to keep this sounding in the data set and just to highlight that our quality control identified it as unusual. We would be very interested to see, if any study can identify a geophysical reason for its unusual deviation from the mean.

9. Line 144: Should this read "within 3% RH" rather than 3%?

Actually, we incorrectly wrote 3%. This should have been 1.5% RH and we corrected the text to that effect.

10. Line 179: Remove "may."

Corrected to " ... may have been damaged ... "

11. Line 259: "a multitude of other meteorological research questions" is mentioned, but it would be good to surmise as to what some of them might be.

Rather than being specific of what members of the OTREC research team is planning, we decided to leave this sentence generic and as advertisement that these data can be used for a many other research questions beyond the initial intent of OTREC. We rephrased it to:

"The temperature, humidity, and wind fields measured by these sondes are being currently used to study aspects of tropical convection in the context of OTREC and provide a unique data set to investigate a multitude of other meteorological research questions beyond the initially proposed research project."